# Reaction of Oat Genotypes to *Fusarium equiseti* (Corda) Sacc. Infection and Mycotoxin Concentrations in Grain

Elżbieta Mielniczuk [1,*], Marcin Wit [2], Elżbieta Patkowska [1,*], Małgorzata Cegiełko [1] and Wojciech Wakuliński [2]

1    Department of Plant Protection, University of Life Sciences in Lublin, Leszczyńskiego 7, 20-069 Lublin, Poland; malgorzata.cegielko@up.lublin.pl

2    Department of Plant Protection, Warsaw University of Life Sciences, Nowoursynowska 159, 02-776 Warsaw, Poland; marcin_wit@sggw.edu.pl (M.W.); wojciech_wakulinski@sggw.edu.pl (W.W.)

*    Correspondence: elzbieta.mielniczuk@up.lublin.pl (E.M.); elzbieta.patkowska@up.lublin.pl (E.P.); Tel.: +48-815248162 (E.M.)

**Abstract:** *Fusarium* head blight and the contamination of cereals with toxic fungal metabolites are particularly important problems in global agriculture. The increasingly frequent isolation of *F. equiseti* from cereal grain and the sparse information in the literature on the harmfulness of this fungus to oat encouraged us to conduct the present research. The aim of the study was to determine the susceptibility of oat genotypes to panicle infection by *F. equiseti* and mycotoxin content in the grain. Field experiments involving 10 oat genotypes were conducted over three years (2015–2017). Oat panicles were inoculated with a conidial suspension of *F. equiseti*, which reduced the kernels yield by 38.34%, the number of kernels per panicle by 31.16% and 1000 kernels weight by 12.66%. *F. equiseti* accumulated type A trichothecenes (T-2 and HT-2 toxins, scirpentriol, diacetoxyscirpenol, T-2 triol, T-2 tetraol) and type B trichothecenes (deoxynivalenol, 3Ac-DON, 15Ac-DON, nivalenol, fusarenone X) in kernels at an average level of 0.0616 and 0.2035 mg·kg$^{-1}$, respectively. The highest susceptibility to scabs caused by *F. equiseti* was found for genotype POB 4901/10, whereas cv. Elegant exhibited the highest resistance to *F. equiseti* in terms of yield reduction after inoculation.

**Keywords:** *Fusarium equiseti*; *Fusarium* panicle blight; mycotoxins; yield reduction; resistance of genotypes to pathogens

## 1. Introduction

*Fusarium* head blight (FHB or scab) is a serious disease of cereals all over the world, including oat. The occurrence of this disease on cereals grown in various climate zones causes significant losses in the grain yield [1–3]. The main pathogens causing head blight are *Fusarium graminearum* (Schwabe), *Fusarium culmorum* (Wm. G. Sm.) Sacc., *Fusarium avenaceum* (Fr.) Sacc. and *Fusarium poae* (Wr.) Peck [1,4–6]. Moreover, cereal ears can also be infected by *F. equiseti* [4,5]. The predominance of a given species in FHB induction is modified, among others, by weather conditions [7,8]. *Fusarium graminearum* develops most optimally in warm and humid weather conditions and it occurs mainly in warmer regions of the world [9]. *F. culmorum*, on the other hand, is the main pathogen of cereals grown mostly in temperate climates, and warm weather with occasional rainfall during flowering is conducive to ear infection [10]. *F. avenaceum* and *F. equiseti* are temperature- and moisture-tolerant species, and very often predominate in cooler climates [2,9,11]. *F. equiseti* is currently included in the *Fusarium incarnatum-equiseti* species complex [12] and has been identified, among others, as the causal agent of *Fusarium* diseases of maize and wheat ears, as well as other cereals grown in Europe and North America [5,11,13,14]. This species has been identified as the cause of *Fusarium* blight of wheat and barley ears and oat panicles grown in Norway, Denmark and Canada [11,14–17]. In addition to weather conditions, the resistance of the cultivated cereal genotypes has a modifying effect on the

severity of *Fusarium* head blight [18]. Oat panicles are usually less affected by *Fusarium* spp. compared to ears of other cereals. However, there are regions where fusarium panicle blight is a serious problem in favorable weather conditions. These include Scandinavia and Canada [14–16].

The infection of ears and panicles by *Fusarium* spp. not only reduces the grain yield, but also poses a risk of cereal grain contamination with compounds toxic to humans and animals [6,19,20]. The factors influencing mycotoxin content in the infected grain are: the climatic and soil conditions, harvest method, grain storage, as well as genetic characteristics of cereal species and cultivars and toxigenic abilities of individual *Fusarium* spp. strains infecting the ears [5,21]. Hence, cultivation of the least susceptible genotypes is an important element of integrated plant protection against toxic pathogens [4]. Optimal fertilization, proper crop rotation and the application of fungicides from different chemical groups are also important elements of integrated cereal protection against *Fusarium* disease development. Most pathogenic fungi can survive in harvest residues; therefore, a properly designed crop rotation, in addition to direct and indirect protection methods (fungicide application at flowering, use of cultivars resistant to *Fusarium* spp.), can significantly reduce *Fusarium* spp. and grain contamination with mycotoxin [22]. Cultivating cereals one after another is a particularly unfavorable form of crop rotation, especially after wheat and maize [3,4,23].

Currently, biological methods play a significant role in plant protection as an element of biocontrol of fungal pathogens by inhibiting their development and reducing the grain mycotoxin content. According to the literature, Good Agricultural Practices (GAP) are the best line of defense for controlling *Fusarium* toxin contamination of cereal and maize grains. However, fluctuations in weather conditions can significantly reduce the effectiveness of plant protection methods against *Fusarium* spp. infection and mycotoxin accumulation in grains [4].

*F. equiseti* produces various quantities of the following substances: nivalenol, fusarenone X, deoxynivalenol and its acetyl derivatives, diacetoxyscirpenol, monoacetoxyscirpenol, T-2 toxin, HT-2 toxin, T-2 triol, T-2 tetraol, scirpentriol, as well as fusarochromanone, zearalenone, butenolide and equisetin [5,24–28]. According to Langseth et al. [24], *F. equiseti* cytotoxicity is comparable to that of the most toxic *F. culmorum* isolates.

The species *F. avenaceum*, *F. culmorum* and *F. poae* are considered to be the main cause of *Fusarium* panicle disease of oat grown in Europe [15,24,29]; however, the increasingly frequent isolation of *F. equiseti* from the grain of this cereal and sparse information in the literature on the harmfulness of this fungus to oat prompted us to conduct the present research. Moreover, oat is widely used in the production of healthy food; therefore the aim of the study was to determine the level of contamination of its grain with toxic *F. equiseti* metabolites, especially in conditions of the epidemic risk of *Fusarium* panicle infection, and to select the genotypes least susceptible to FHB.

## 2. Materials and Methods

### 2.1. Field Experiment

Susceptibility of panicles of selected 10 oat genotypes (cultivars: Agent, Elegant, Denar, Kozak, Romulus, lines: DC 06011-8, DC 14-8, POB 961-1344/13, POB 4109/10 and POB 6020/10) to *F. equiseti* No. O-020 infection was determined based on a field experiment with artificial panicle infection during the flowering stage. Oat cultivars and lines used in inoculation experiments were originally developed and introduced in Poland. The seed material was submitted to phytopathological testing by the breeders: Danko Plant Breeding, Sp. z o. o., Strzelce Plant Breeding, Sp. z o. o., Plant Breeding and Acclimatization Institute (IHAR)—National Research Institute Group and Małopolska Plant Breeding, Sp. z o. o. as part of a research project. Each oat genotype whose panicles were artificially contaminated grew in a separate 10-m$^2$ plot. The plots were located in south-eastern Poland, near Zamość.

2.1.1. Infectious Material

*F. equiseti* strain No. O-020 was derived from the culture collection of the Department of Phytopathology and Mycology, Department of Plant Protection, University of Life Sciences in Lublin, and was isolated from oat kernels obtained from the panicles with *Fusarium* blight symptoms. The fungus was isolated from kernels using the plate method on a mineral medium with the following composition: 38 g of sucrose, 0.7 g of $NH_4NO_3$, 0.3 g of $KH_2PO_4$, 0.3 g of $MgSO_4 \times 7\,H_2O$, trace amount of $FeCl_3 \times 6\,H_2O$, trace amount of $ZnSO_4 \times 7\,H_2O$, trace amount of $CuSO_4 \times 7\,H_2O$, trace amount of $MnSO_4 \times 5\,H_2O$ and 20 g of agar. Water was added to a final volume of 1000 mL. *F. equiseti* was identified by the classical method based on the observation of morphological structures formed on standard media (potato dextrose agar—PDA, carnation leaf agar—CLA and selective medium SNA (synthetic nutrient agar), according to the keys and monographs of Nelson et al. [30] and Leslie and Summerell [31].

The fungus grew rapidly on PDA medium, with the colony reaching the diameter of 5.8 cm after only 4 days. The aerial mycelium was abundant, flocculent at first and then woolly, yellow-beige in color. The reverse of the colony was brown in the center and slightly yellow in the periphery. Macroconidia were sickle-shaped, dorsoventrally curved with an elongated basal cell ending in a prominent long foot; the apical cell was also elongated, tapered, slightly curved. Spores mostly had 4–6 cells and formed on monophialide. Macroconidia formed in orange sporodochia in the central part of the colony. Microconidia were absent. The tested *F. equiseti* strain formed thick-walled, spherical, rough, yellow-brown, numerous chlamydospores in mycelial hyphae, which occurred singly, in pairs, or were produced in clumps or chains. The morphological structures of the tested *F. equiseti* strain were consistent with the description provided by Kosiak et al. [28] and Leslie and Summerell [31].

The profile of mycotoxins produced by the analyzed strain of the fungus was determined by high-performance gas chromatography at the Department of Chemistry, Poznań University of Life Sciences, Poland. *F. equiseti* No. O-020 produced trichothecenes compounds of groups A (T-2 toxin, HT- 2 toxin, scirpentriol (STO), T-2 tetraol, diacetoxyscirpenol (DAS), T-2 triol) and B (deoxynivalenol (DON), 3-Ac DON, 15-AcDON and nivalenol (NIV). The results of these studies were consistent with the results of other authors [24,26,27]. Before the field tests, the pathogenicity of the analyzed strain was assessed using the method of Mishra and Behr [32] based on the evaluation of the germination capacity of oat kernels of the cultivar Bingo. The fungus reduced oat kernels germination to 11%, while 92% kernels germination was recorded in the control combination.

The *F. equiseti* No. O-020 infectious material was a macroconidia suspension with a density of $5 \times 10^5$ spores mL$^{-1}$. The infectious mixture was prepared as in the research described by Kiecana and co-authors [33]: the growing medium (1:1) was composed of water extract from 0.5 kg of oat leaves and a liquid selective medium—SNA without the addition of agar, autoclaved for 1 h at 121 °C and 1 atm. The medium was inoculated with the mycelium of a two-week-old culture of *F. equiseti* strain No. O-020 and incubated for two weeks at 18–20 °C with a 12-h period of natural light. After incubation, the inoculum was stirred for 10 min, filtered through a cheesecloth, and the supernatant of the conidial suspension ($5 \times 10^5$ spores mL$^{-1}$) was used for the inoculation of panicles in field conditions.

2.1.2. Panicle Inoculation

Panicle inoculation was carried out with a garden sprayer using 4 mL of the infectious material per panicle. One hundred panicles of each oat genotype ($4 \times 25$ panicles per replicate) were inoculated with *F. equiseti* 4 days after anthesis of a minimum of 50% of plants (on 20–23 June 2015, 25–27 June 2016 and June 30–July 2 2017). The panicles of the control combination (non-inoculated group) were sprayed with distilled water only. After inoculation, the panicles were covered with plastic bags, and thus the infectious material was protected against air currents and drying for 24 h [33].

The inoculated and control panicles were cut at the stage of full grain maturity (harvest dates: 27 July 2015, 2 August 2016 and 5 August 2017), the kernels were separated and the following parameters were determined: the number of kernels per panicle (number of kernels was determined in 40 panicles—4 replicates × 10 panicles), the kernels yield of 40 panicles (4 × 10 panicles) and 1000 kernels weight (TKW). The results obtained from the panicle inoculation experiment were compared to the control.

### 2.1.3. Fungus Re-Isolation

Fungus re-isolation from the kernels of all experimental combinations was conducted to complete Koch's postulates. Re-isolation of fungi from kernels obtained from panicles inoculated with *F. equiseti* was performed using the plate method. Mineral medium (medium with the composition given above), was used to isolate the fungi colonizing the oat kernels. For each genotype, 50 kernels from the combination of the panicle inoculation experiment were analyzed. Fungi of the genus *Fusarium* were determined on standard media as in the identification of the *F. equiseti* strain used for panicle inoculation. Colonies of other fungi were determined according to the keys and monographs presented by Mielniczuk and Cegiełko [34].

### 2.2. Chemical Analyses

Grain samples from oat panicles inoculated with *F. equiseti* were analyzed for the presence of trichothecenes according to Perkowski et al. [35]. Grain samples (10 g) of each oat genotype were ground and subsequently extracted with acetonitrile/water (82:18) and purified on a charcoal column [Celite 545/charcoal Draco G/60/activated alumina neutral 4:3:4 ($w/w/w$)]. Mycotoxin content was analyzed in four replications. Type A trichothecenes were analyzed as trifluoroacetyl derivatives. The amount of 100 µL trifluoroacetic acid anhydride was added to the dried sample. After 20 min the reacting substance was evaporated to dryness under nitrogen. The residue was dissolved in 500 µL of isooctane and 1 µL was injected onto a gas chromatograph-mass spectrometer. Type B trichothecenes were analyzed as trimethylsilyl derivatives. A 100-µL volume of a TMSI/TMCS mixture (trimethylsilyl imidazole/trimethylchlorosilane, $v/v$ 100/1) was added to the dried extract. After 10 min, 500 µL of isooctane was added and the reaction was quenched with 1 mL of water. The isooctane layer was used for the analysis and 1 µL of the sample was injected on a GC/MS system. Chromatographic separation and analysis were carried out separately using a gas chromatograph (Hewlett Packard 6890) on a capillary column (HP-5MS, 0.25 mm × 30 m capillary column) coupled to a mass detector (Hewlett Packard 5972 A). The analysis was performed in the selected ion monitoring (SIM) mode. For type A trichothecenes, those were: STO 456 and 555; T-2 tetraol 455 and 568; T-2 triol 455 and 569; DAS 402 and 374; HT-2 455 and 327; T-2 327 and 401. For type B trichothecenes, those were: DON 103 and 512; 3-AcDON 117 and 482; 15-AcDON 193 and 482; FUS 103 and 570; NIV 191 and 600. The helium flow rate was 0.7 cm$^3$·min$^{-1}$. A full mass range (from 100 to 700 amu) analysis was performed to confirm the presence of the determined toxins in the samples. The obtained results were processed using the Chem Station program. The recovery percentages for the analyzed toxins were: T-2—86 ± 3.8%; T-2 tetraol—88 ± 4.0%; HT-2—91 ± 3.3%; DAS—84 ± 4.6%; DON—84 ± 3.8%; 3AcDON—78 ± 4.8%; AcDON—74 ± 2.2% and NIV—81 ± 3.8%. The detection limit for the analyzed toxins was 0.001 mg·kg$^{-1}$.

### 2.3. Statistical Analysis

Statistical calculations of the results obtained in the field experiment involving oat panicle inoculation with *F. equiseti* during flowering, and the results of the chemical analysis for the presence of toxic metabolites were performed using the statistical Statistica software, version 13 (StatSoft Polska, Kraków, Poland) and ARSTAT, developed at the Department of Applied Mathematics and Computer Science at the University of Life Sciences in Lublin. An analysis of variance was performed to determine the effect of panicle inoculation with

the tested fungal strain, where the dependent variables were the absolute values of the yield structure elements (kernels yield, number of kernels per panicle—NKP and 1000 kernels weight—TKW). The Tukey test was used to compare the means for individual cultivars at two factor levels: inoculation and control. An analysis of variance was also performed to assess the reduction of the tested yield components (RYIELD, RNKP, RTKW). For the purpose of statistical analysis, the percentage data were transformed according to Bliss formula (f(x) = Arcsin $\sqrt{x}$). Pearson correlation coefficients between the components of the kernels yield reduction were also calculated. All the hypotheses were verified with $p \leq 0.05$.

### 2.4. Weather Conditions

Data on temperature and precipitation during oat vegetation seasons in 2015–2017 were obtained from the Meteorological Station in Zamość, belonging to the Department of Environmental Protection and Development of the Faculty of Agricultural Sciences in Zamość, University of Life Sciences in Lublin and the Internet publications of the Institute of Meteorology and Water Management (www.imgw.pl/klimat) (accessed on 1 December 2021).

## 3. Results

The method of panicle inoculation with a suspension of *F. equiseti* No. O-020 macroconidia turned out to be effective, because the panicles showed disease symptoms and etiological signs characteristic of scab, with a higher number of spikelets affected in the panicle than under natural infection conditions. Orange sporodochia of the fungus were observed on the chaff of spikelets (Figure 1). The affected spikelets whitened prematurely. The kernels from the infected panicles were smaller, shriveled and discolored (gray) compared to the control, sometimes covered with a layer of mycelium. Some spikelets in the inoculated panicles remained infertile. In contrast, the panicles from the control combination showed normal development and color, with well-developed kernels (Figure 2).

The statistical analysis of the mean results obtained over three study years (2015–2017) showed that panicle inoculation with the strain *F. equiseti* No. O-020 during flowering significantly influenced the number of kernels per panicle, the kernels yield of 10 panicles and 1000-kernel weight in the analyzed genotypes of this cereal compared to control, the exceptions being the number of kernels per panicle in the Elegant cultivar (Table 1). The effects of *F. equiseti* on the yield structure of oat in 2015–2017 are presented in supplementary files (Supplementary Tables S1–S3). The harmful effect of the fungus on oat plants was observed in all three growing seasons. The mean number of kernels, kernels yield and 1000 kernels weight were significantly reduced compared to non-infected plants in 2015 (Supplementary Table S1), 2016 (Supplementary Table S2) and 2017 (Supplementary Table S3). The significant of interaction genotype × year was found, indicating the role of the environment on the oat genotype response. In the particular years of the study some genotypes reaction was similar to the control and the reduction of yield structure parameters nonsignificant.

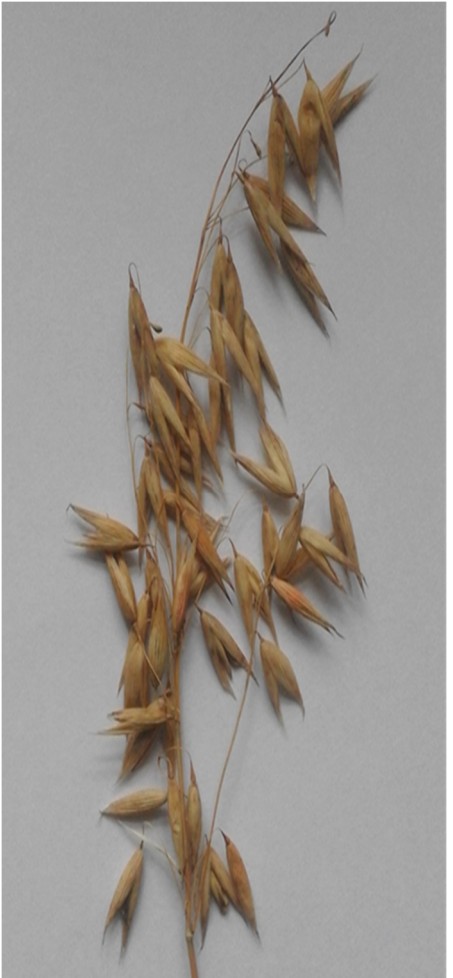
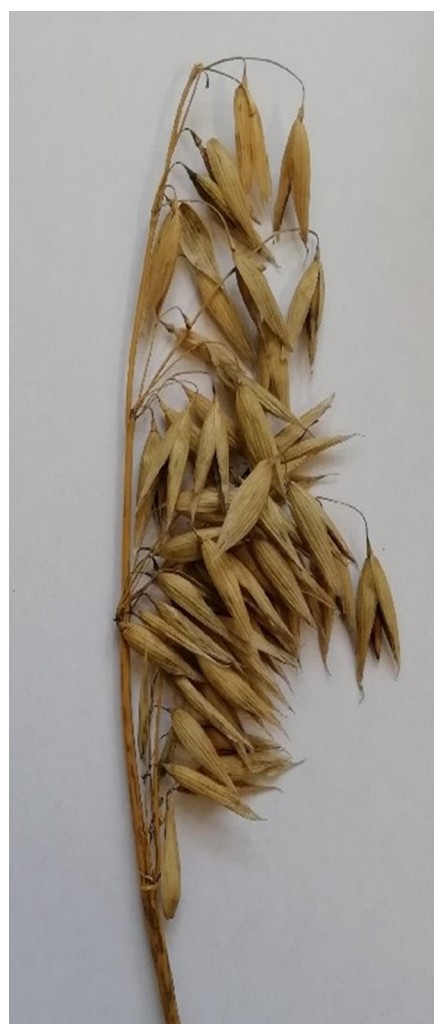

**Figure 1.** Oat panicles after inoculation with *F. equiseti* (**left**) and control (**right**).

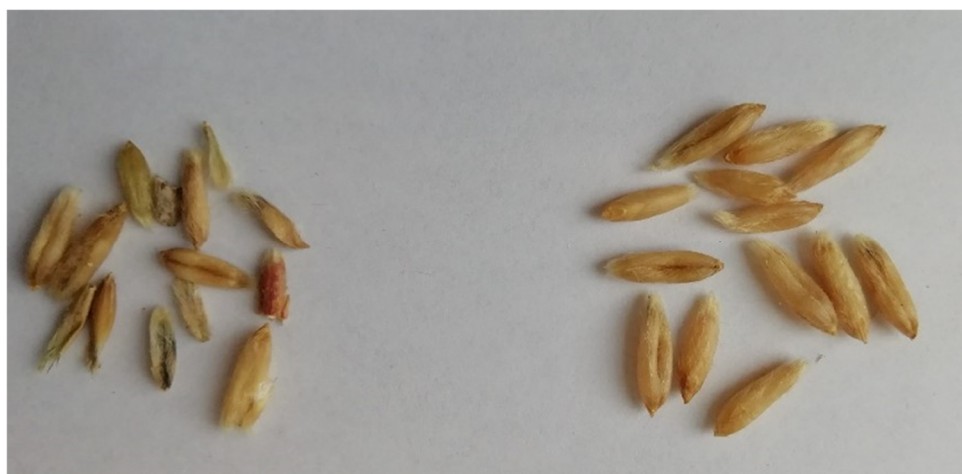

**Figure 2.** Oat kernels (without chaff) from panicle after inoculation with *F. equiseti* (**left**) and kernels from control panicles (without inoculation) (**right**).

**Table 1.** Effect of oat panicle inoculation with *F. equiseti* No O-020 on the number of kernels per panicle [1], kernels yield per 10 panicles and 1000 kernels weight; average over three study years (2015–2017).

| Genotypes | Elements of Yield Structure | | | | | |
|---|---|---|---|---|---|---|
| | Number of Kernels | | Kernels Yield (g) | | 1000 Kernels Weight (g) | |
| | *F. eq.* Inoculated | Control | *F. eq.* Inoculated | Control | *F. eq.* Inoculated | Control |
| Agent | 48.70 * | 71.57 | 18.31 * | 27.89 | 37.57 * | 39.86 |
| Elegant | 61.48 | 69.12 | 19.42 * | 25.14 | 30.66 * | 36.26 |
| Denar | 45.38 * | 72.65 | 14.30 * | 24.37 | 30.84 * | 35.60 |
| Kozak | 59.24 * | 79.58 | 18.21 * | 29.38 | 33.27 * | 38.17 |
| Romulus | 61.95 * | 74.75 | 19.35 * | 28.37 | 30.32 * | 38.24 |
| DC06011-8 | 58.62 * | 91.30 | 18.53 * | 32.11 | 32.73 * | 37.02 |
| DC 14-8 | 54.21 * | 73.50 | 15.93 * | 24.58 | 30.63 * | 33.99 |
| POB 4109/10 | 35.07 * | 69.47 | 10.87 * | 24.77 | 24.60 * | 33.07 |
| POB 6020/10 | 60.17 * | 83.63 | 18.71 * | 27.09 | 25.71 * | 30.08 |
| POB 961-1344/13 | 48.87 * | 83.84 | 14.45 * | 27.94 | 23.99 * | 34.62 |
| Average | 53.37 * | 76.94 | 16.81 * | 27.16 | 30.03 * | 35.69 |

[1] Number of kernels per panicle were analyzed in 40 panicles (4 × 10) in each study year; * Values differ significantly compared to control at $p \leq 0.05$.

The comparative analysis showed differences between oat genotypes in terms of the reduction of the analyzed yield structure elements. On average, over 3 years of research (2015–2017), the smallest reduction in the number of kernels in the panicle and the kernels yield of 10 panicles was recorded for the cultivar Elegant: 13.63% and 24.35%, respectively, compared to control (Figures 3 and 4). The average reduction in 1000 kernels weight (TKW) for this cultivar was 11.87% (Figure 5). In individual study years, the reduction in the kernels number per panicle in the cultivar Elegant ranged from 0.49% (2017) to 36.60% (2016), while the reduction in the kernels yield was from 7.35% (2017) to 40.50% (2016), and TKW from 6.50% (2016) to 22.30% (2015) (Figures 3–5). Significantly the highest average reduction in the number of kernels per panicle and the kernels yield of 10 panicles, over 3 years of research, was found in breeding line POB 4109/10: 51.73% (from 32.60% in 2015 to 65.90% in 2016) and 57.67% (from 44.80% in 2015 to 64.60% in 2016), respectively (Figures 3 and 4). The greatest reduction in 1000 kernels weight was recorded for cv. Romulus—20.75% (from 0.95% in 2017 to 34.7% in 2016) (Figure 5).

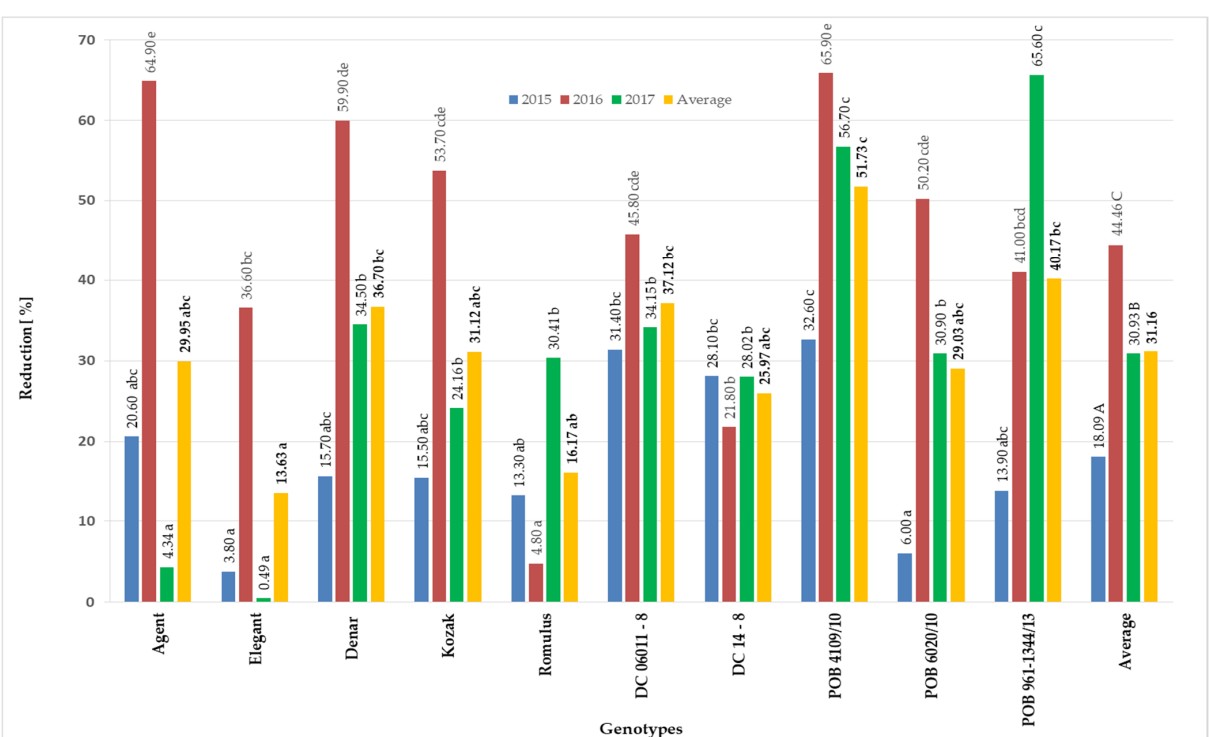

**Figure 3.** Reduction in the kernels number per panicle after inoculation with *F. equiseti* compared to control. The values obtained within an individual year for all genotypes, marked with the same letter do not differ significantly at $p \leq 0.05$. The average values obtained for all genotypes over the three years of study (yellow columns), marked with the same letter, do not differ significantly at $p \leq 0.05$.

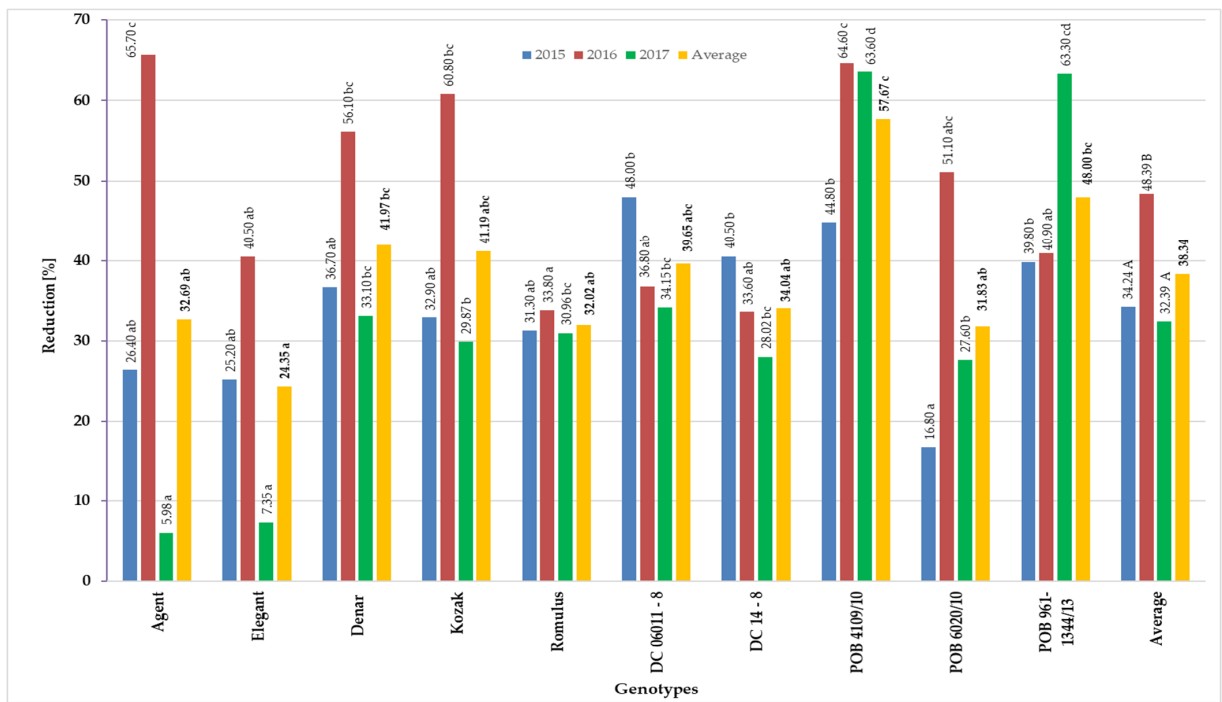

**Figure 4.** Kernels yield reduction after panicle inoculation with *F. equiseti* compared to control. The values obtained within an individual year for all genotypes, marked with the same letter do not differ significantly at $p \leq 0.05$. The average values obtained for all genotypes over the three years of study (yellow columns), marked with the same letter, do not differ significantly at $p \leq 0.05$.

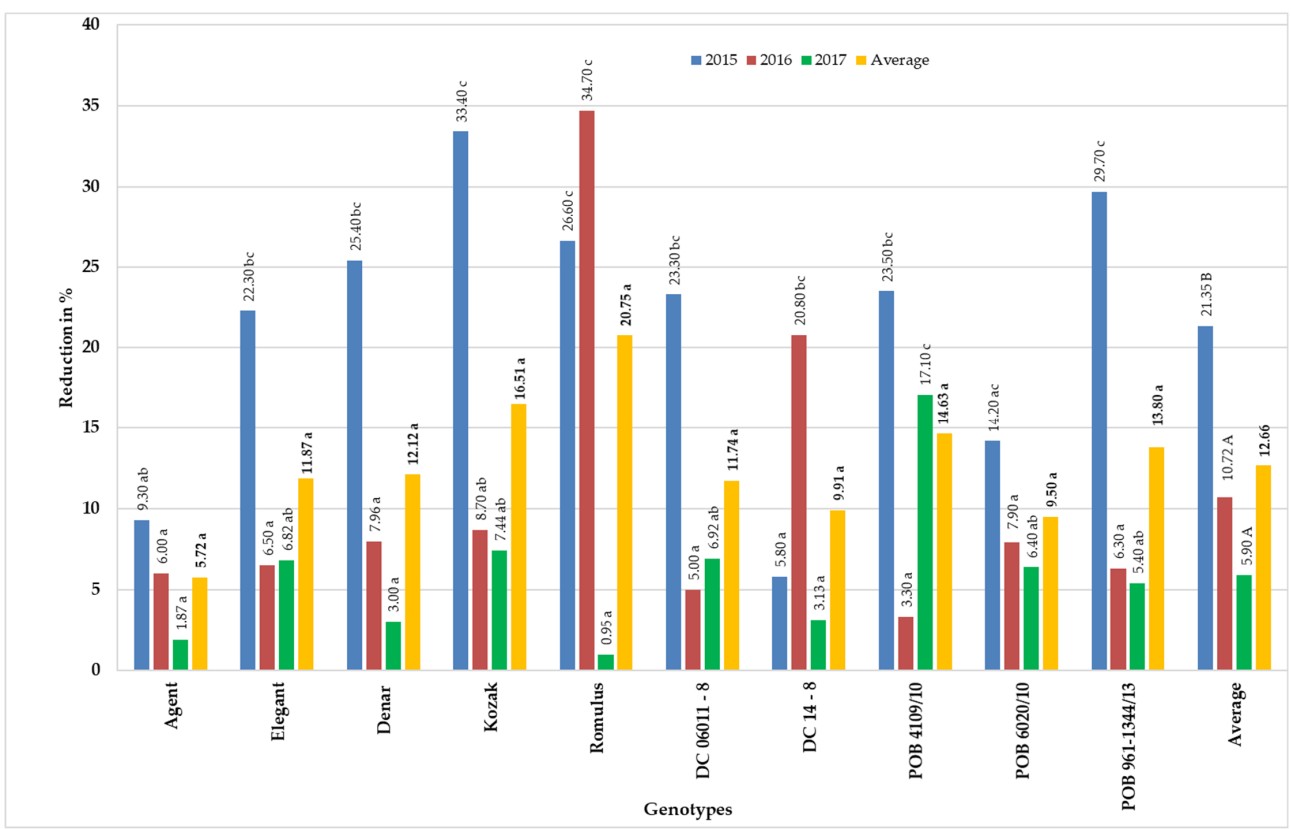

**Figure 5.** Reduction of 1000 kernels weight after panicle inoculation with *F. equiseti*, compared to control. The values obtained within an individual year for all genotypes, marked with the same letter do not differ significantly at $p \leq 0.05$. The average values obtained for all genotypes over the three years of study (yellow columns), marked with the same letter, do not differ significantly at $p \leq 0.05$.

Panicle inoculation with *F. equiseti* No. O-020 macroconidia, of all tested oat genotypes from the years 2015–2017, resulted in a different extent of the kernels number reduction per panicle, the kernels yield and TKW; however, there were no statistically significant differences in the kernels yield between 2015 and 2017, and in 1000 kernels weight (TKW) between 2016 and 2017 (Figures 3–5). The lowest decrease in the number of kernels per panicle (18.09%) was recorded in 2015, and the highest in 2016 (44.46%) (Figure 3). The lowest average kernels yield loss was found in 2017 and 2015, 32.39% and 34.24%, respectively, and the highest (48.39%) in 2016 (Figure 4). As regards TKW, the smallest reduction was recorded in 2017 (5.90%), and the largest (21.35%) in 2015 (Figure 5).

In addition, the statistical analysis showed that the kernels yield reduction was most significantly related to the decrease in the kernel number per panicle (r = 0.821), while it was less significantly associated with lower TKW (r = 0.091). There was also a negative correlation between the reduction in the number of kernels per panicle and 1000 kernels weight (r = −0.338) (Table 2).

**Table 2.** Correlation coefficients between kernels yield reduction (RKY), number of kernels per panicle reduction (RNKP) and 1000 kernels weight reduction (RTKW), on average over 3 years of research (2015–2017).

| Variabiles Analyzed Parametrs | RKY | RNKP | RTKW |
|---|---|---|---|
| RKY | 1 | 0.821 * | 0.091 |
| RNKP | 0.821 * | 1 | −0.338 * |
| RTKW | 0.091 | −0.338 * | 1 |

* values display a significant relationship, at significance level α = 0.05.

Re-isolation of the fungus from inoculated kernels carried out according to Koch's postulates confirmed their colonization by *F. equiseti* (Table 3).

**Table 3.** Fungi isolated from kernels obtained from panicles inoculated with *F. equiseti* in 2015–2017.

| Fungus Species | Number of Isolates in Each Study Year | | | Total Number of Isolates |
| --- | --- | --- | --- | --- |
| | **2015** | **2016** | **2017** | |
| *Alternaria alternata* (Fr.) Keissler | 5 | 8 | 10 | 23 |
| *Aspergillus niger* Link | 0 | 2 | 0 | 2 |
| *Cladosporium cladosporioides* (Fresen.) G.A. de Vries | 1 | 6 | 4 | 11 |
| *Drechslera avenae* (Edima) Scharif | 2 | 0 | 0 | 2 |
| *Epicoccum nigrum* Link ex Link | 34 | 29 | 12 | 75 |
| *Fusarium equiseti* (Fr.) Sacc. | 433 | 396 | 401 | 1230 |
| *Fusarium poae* (Peck.) Wr. | 1 | 2 | 1 | 4 |
| *Mucor hiemalis* Wehmer | 2 | 0 | 0 | 2 |
| *Penicillium* spp. | 9 | 2 | 5 | 16 |
| *Stemphylium botryosum* Wallr. | 0 | 2 | 1 | 3 |
| Total | 487 | 447 | 434 | 1368 |

### 3.1. Mycotoxin Contents in Grain Obtained from Panicles Inoculated with F. equiseti at the Flowering Stage

The chemical analysis of kernels from panicles artificially infected from 2015–2017 showed the presence of types A and B trichothecene compounds (Table 4). The average total of type A trichothecenes over 3 years of testing of all genotypes ranged from 0.0442 mg·kg$^{-1}$ (cv. Agent) to 0.0913 mg·kg$^{-1}$ (cv. Romulus) (Table 4).

**Table 4.** Content of type A trichothecenes (T-2/HT-2—toxin, diacetoxyscirpenol—DAS, scirpentriol—STO, T-2 triol, T-2 tetraol) in oat kernels after panicle inoculation with *F. equiseti*.

| Genotypes | Average Mycotoxin Content over 3 Years of Research [mg·kg$^{-1}$] | | | | | | Sum of Group A Trichothecenes |
| --- | --- | --- | --- | --- | --- | --- | --- |
| | **T-2** | **HT-2** | **DAS** | **STO** | **T-2 Triol** | **T-2 Tetraol** | |
| Agent | 0.0000 | 0.0043 | 0.0023 | 0.0334 | 0.0023 | 0.0020 | 0.0442 a |
| Elegant | 0.0009 | 0.0128 | 0.0056 | 0.0299 | 0.0017 | 0.0029 | 0.0538 b |
| Denar | 0.0050 | 0.0083 | 0.0014 | 0.0308 | 0.0010 | 0.0082 | 0.0547 b |
| Kozak | 0.0000 | 0.0055 | 0.0247 | 0.0290 | 0.0003 | 0.0014 | 0.0610 b |
| Romulus | 0.0007 | 0.0282 | 0.0162 | 0.0325 | 0.0005 | 0.0132 | 0.0913 c |
| DC06011-8 | 0.0000 | 0.0067 | 0.0029 | 0.0298 | 0.0003 | 0.0050 | 0.0448 a |
| DC 14-8 | 0.0003 | 0.0100 | 0.0009 | 0.0289 | 0.0002 | 0.0067 | 0.0470 a |
| POB 4109/10 | 0.0147 | 0.0113 | 0.0033 | 0.0391 | 0.0010 | 0.0179 | 0.0872 c |
| POB 6020/10 | 0.0020 | 0.0075 | 0.0010 | 0.0305 | 0.0007 | 0.0091 | 0.0507 b |
| POB 961-1344/13 | 0.0003 | 0.0156 | 0.0015 | 0.0429 | 0.0010 | 0.0213 | 0.0827 c |
| Average mycotoxin content | 0.0024 | 0.0110 | 0.0059 | 0.0327 | 0.0009 | 0.0088 | 0.0616 |

Values in columns marked with the same letter do not differ significantly at $p \leq 0.05$.

The average sums of type A trichothecene compounds in the grain of all oat genotypes in 2015, 2016, and 2017 were 0.0997, 0.0210 and 0.0643 mg·kg$^{-1}$, respectively (Table 5). Among type A trichothecenes, scirpentriol had the highest concentration—0.0327 mg·kg$^{-1}$ in oat grain, while T-2 triol the lowest—0.0009 mg·kg$^{-1}$. In addition, grain from panicles inoculated with *F. equiseti* was contaminated with HT-2 toxin, diacetoxyscirpenol and T-2 tetraol (Tables 4 and 5).

The average value over 3 years of testing the total of type B trichothecenes for individual genotypes ranged from 0.0945 mg·kg$^{-1}$ (POB 4109/10) to 0.3574 mg·kg$^{-1}$ (Kozak) (Table 6). The average sums of these metabolites in the grain of all oat genotypes in 2015, 2016, and 2017 were 0.1708, 0.1201 µg·kg$^{-1}$ and 0.3192 mg·kg$^{-1}$, respectively. Over the three study years, nivalenol was found in the highest amounts in oat grain obtained from panicles inoculated with *F. equiseti* (on average 0.1362 mg·kg$^{-1}$—from 0.0692 in 2015 to

0.2455 mg·kg$^{-1}$ in 2017), while 15-AcDON was determined in the lowest quantities. Kernels from panicles inoculated with *F. equiseti* were also contaminated with deoxynivalenol, 3-Ac DON and FUS X (Table 7).

**Table 5.** Content of type A trichothecenes (T-2/HT-2—toxin, diacetoxyscirpenol—DAS, scirpentriol—STO, T-2 triol, T-2 tetraol) in oat kernels after panicle inoculation with *F. equiseti*.

| Years | Mycotoxin Contents [mg·kg$^{-1}$], Average for All Genotypes | | | | | | |
| | T-2 toxin | HT-2 toxin | DAS | STO | T-2 Triol | T-2 Tetraol | Sum of Group A Trichothecenes |
|---|---|---|---|---|---|---|---|
| 2015 | 0.0001 | 0.0138 | 0.0003 | 0.0798 | 0.0001 | 0.0056 | 0.0997 c |
| 2016 | 0.0003 | 0.0075 | 0.0022 | 0.005 | 0.0002 | 0.0058 | 0.0210 a |
| 2017 | 0.0068 | 0.0118 | 0.0151 | 0.0133 | 0.0024 | 0.0149 | 0.0643 b |
| Average mycotoxin content | 0.0024 | 0.0110 | 0.0059 | 0.0327 | 0.0009 | 0.0088 | 0.0616 |

Values in columns marked with the same letter do not differ significantly at $p \leq 0.05$.

**Table 6.** Content of type B trichothecenes (deoxynivalenol—DON, 3-ACDON, 15-AcDON, nivalenol—NIV, fusarenone X—FUS X) in oat kernels after panicle inoculation with *F. equiseti*.

| Genotypes | Average Mycotoxin Contents over 3 Years of Research [mg·kg$^{-1}$] | | | | | |
| | DON | 3-ACDON | 15-AcDON | NIV | FUS X | Sum of Group B Trichothecenes |
|---|---|---|---|---|---|---|
| Agent | 0.0227 | 0.0073 | 0.0067 | 0.1860 | 0.0094 | 0.2322 b |
| Elegant | 0.0160 | 0.0077 | 0.0130 | 0.1793 | 0.0105 | 0.2264 b |
| Denar | 0.0177 | 0.0157 | 0.0055 | 0.0634 | 0.0078 | 0.1101 a |
| Kozak | 0.0495 | 0.0113 | 0.0053 | 0.2827 | 0.0085 | 0.3574 c |
| Romulus | 0.0365 | 0.0173 | 0.0198 | 0.2564 | 0.0084 | 0.3384 c |
| DC06011-8 | 0.0277 | 0.0087 | 0.0105 | 0.1456 | 0.0301 | 0.2226 b |
| DC 14-8 | 0.0209 | 0.0770 | 0.0070 | 0.0859 | 0.0106 | 0.2014 b |
| POB 4109/10 | 0.0116 | 0.0153 | 0.0058 | 0.0537 | 0.0082 | 0.0945 a |
| POB 6020/10 | 0.0517 | 0.0150 | 0.0055 | 0.0343 | 0.0081 | 0.1146 a |
| POB 961-1344/13 | 0.0343 | 0.0153 | 0.0053 | 0.0745 | 0.0081 | 0.1376 a |
| Average mycotoxin content | 0.0287 | 0.0191 | 0.0084 | 0.1362 | 0.0110 | 0.2034 |

Values in columns marked with the same letter do not differ significantly at $p \leq 0.05$.

**Table 7.** Content of type B trichothecenes (deoxynivalenol—DON, 3-ACDON, 15-AcDON, nivalenol—NIV, fusarenon X—FUS X) in oat kernels after panicle inoculation with *F. equiseti*.

| Years | Mycotoxin Contents [mg·kg$^{-1}$] | | | | | |
| | DON | 3-Ac DON | 15-Ac DON | NIV | FUS X | Sum of Group B Trichothecenes |
|---|---|---|---|---|---|---|
| 2015 | 0.0194 | 0.0421 | 0.0159 | 0.0692 | 0.0242 | 0.1708 a |
| 2016 | 0.0190 | 0.0000 | 0.0032 | 0.0938 | 0.0040 | 0.1201 a |
| 2017 | 0.0477 | 0.0151 | 0.0062 | 0.2455 | 0.0047 | 0.3192 b |
| Average | 0.0287 | 0.0191 | 0.0084 | 0.1362 | 0.0110 | 0.2034 |

Values in columns marked with the same letter do not differ significantly at $p \leq 0.05$.

*3.2. Weather Conditions*

In Zamość, the 2015 growing season was characterized by temperatures higher than the long-term average in June, July, and August by +0.2 °C up to +3.7 °C. The percentage of normal rainfall, as compared to the long-term average, was higher only in May by 75.3%. The oat growing season of 2016 was characterized by higher temperatures (0.5 to 1.7 °C) compared to the long-term average. A greater amount of rainfall was also recorded in the months of April, May and July 2016 compared to the long-term average. On the other hand, the 2017 oat growing season was characterized by temperatures higher than the long-term average in May-August, by 0.2 to 1.7 °C. It was also observed that precipitation in this year of the study was lower than the long-term average (Table 8).

**Table 8.** Weather conditions in Zamość, (Lublin region) during 2015–2017 oat growing seasons.

| Month | Air Temperature [°C] | | | |
| --- | --- | --- | --- | --- |
| | Average for the Years 1971–2005 | 2015 | 2016 | 2017 |
| April | 7.9 | 7.5 | 9.1 | 7.5 |
| May | 14.1 | 13.0 | 14.7 | 14.31 |
| June | 16.8 | 17.0 | 18.5 | 17.5 |
| July | 18.4 | 20.0 | 19.8 | 18.7 |
| August | 17.8 | 21.5 | 18.3 | 19.5 |
| | Rainfall [mm] | | | |
| | Average for the Years 1971–2005 | 2015 | 2016 | 2017 |
| April | 44.1 | 40.0 | 90.0 | 35.0 |
| May | 65.5 | 114.8 | 85.0 | 54.0 |
| June | 78.9 | 55.0 | 48.0 | 45.0 |
| July | 98.4 | 64.8 | 99.0 | 83.0 |
| August | 54.3 | 4.8 | 50.0 | 36.0 |

## 4. Discussion

Various methods of ear inoculation are used in phytopathological experiments to determine genotype susceptibility to infection by *Fusarium* spp. [14,36–40]. The current study applied the method of spraying panicles with a suspension of *F. equiseti* macroconidia (after Kiecana et al. [33]) as it was found to be most consistent with natural infection. Considering the variable virulence of *Fusarium* spp. strains [41–44], the study used *F. equiseti* strain No. O-020, whose pathogenicity was verified using the method described by Mishra and Behr [32]. McCallum and Tekauz [45] and Vančo et al. [46] reported that ears sprayed with a suspension of *F. graminearum* and *F. culmorum* macroconidia yielded a higher number of infected kernels per ear than those in which single spikelets were spot-inoculated.

The conducted research showed that the inoculation of panicles with *F. equiseti* strain No. O-020 during the flowering stage decreased to a large extent the number of kernels per panicle, while their weight was reduced to a lesser extent. The percentage of infected panicles in the canopy determines yield losses in the cultivated area. The kernels yield from panicles artificially infected with *F. equiseti* was lower (compared to control) by an average of 39.5%. The conducted research showed that under conditions of artificial panicle infection *F. equiseti* exerted a similar effect on the reduction of the oat kernels yield to that of *F. avenaceum* and *F. crookwellense* in 1996–1998, as well as *F. poae* in 2008 [33,47,48]. In contrast, *F. culmorum* proved to be more harmful to oat panicles [49]. *F. equiseti* was less pathogenic to spring wheat ears than *F. avenaceum*, *F. culmorum* and *F. graminearum* under controlled conditions [13]. In some regions of South Africa this species was found to induce *Fusarium* head blight more frequently than *F. graminearum*, which was considered the main cause of FHB in that country [50]. *F. equiseti* was recognized as one of the factors deteriorating the quality of cereal grains, including oat grown in Norway [11,24,28].

*F. equiseti* was also isolated from the roots and stem bases of oat [51]. Moreover, this fungus turned out to be the main cause of winter wheat root necrosis and was found to be significantly involved in damaging stem bases of this plant cultivated in the conditions of north-eastern Poland [52].

The results of chemical studies concerning the profile and quantity of mycotoxins produced by *F. equiseti* confirmed the previous findings that this fungus was capable of producing type A and B trichothecenes [5,25,28]. The composition of mycotoxins produced in kernels from oat panicles artificially infected with *F. equiseti* was similar to the metabolite profile of this fungus obtained in laboratory conditions [24,26,28]. The presence of T-2 and HT-2 toxins, diaceotoxyscirpenol, T-2 triol, T-2 tetraol, scirpentriol, as well as

deoxynivalenol, 3Ac-DON and nivalenol was found in grain samples of the studied genotypes. However, neosolaniol, monoacetoxyscirpenol or fuzarenone X and zearalenone—i.e., metabolites considered to be characteristic of the mycotoxin profile produced by different *F. equiseti* isolates—were not detected in oat kernels infected with the analyzed *F. equiseti* strain [25,28,53]. The concentration of mycotoxins in the grain of oat genotypes was lower compared to the same metabolites produced by different *F. equiseti* isolates under laboratory conditions [28]. In the present study, *F. equiseti* strain No. O-020 produced nivalenol in the highest amounts, and the average concentration of this metabolite was 0.1362 mg·kg$^{-1}$,which was lower than in oat grain from panicles artificially infected with *F. crookwellese* and *F. culmorum* [48,49].

The level of oat grain contamination with HT-2 toxin, as a result of artificial panicle infection with *F. equiseti*, was also higher compared to panicles inoculated with *F. poae* No. 35 in 2008 [47]. However, *F. poae* strain No. 35 produced higher amounts of DAS, T-2 tetraol and scirpentriol in oat grain compared to the analyzed *F. equiseti* strain [47].

Similar to other studies on cereals [33,37,40,43,54], the analyzed genotypes differed in their susceptibility to panicle infection by *F. equiseti* at the flowering stage. The results of field experiments demonstrated a high variability in the reaction of the tested cultivars to panicle infection by the fungal strain in individual years of the study, which indicated a significant influence of the genotypic-environmental interaction on the development of *Fusarium* head blight. High temperatures and precipitation in July 2016, directly after panicle inoculation, favored the development of *Fusarium* head blight and grain contamination with mycotoxins. Similar relationships in wheat and triticale cultivars were demonstrated by Chełkowski and co-authors [54], Vogelgsang and co-authors [43] and Weber and Kita [23]. On the other hand, in the season of 2017, which showed the lowest grain reduction compared to the control, a small amount of rainfall and the relatively lowest temperature in June and July—i.e., during flowering and during grain formation—were recorded. The conditions of 2017 were therefore not conducive to the development of FHB caused by *F. equiseti*.

The present study did not show a positive relationship between trichothecene content and kernels yield reduction as a result of panicle inoculation with *F. equiseti*, similar to the analyses of wheat and barley infected by *F. culmorum* [55,56]. The relationship between mycotoxin content and yield reduction seems to be dependent on the type of cultivar resistance to FHB [18,57].

## 5. Conclusions

The conducted research confirmed the harmfulness of *F. equiseti* to oat.

Assuming kernels yield reduction as a result of panicle inoculation with *F. equiseti* as the evaluation criterion, the cultivar Elegant and breeding line POB 6020/10 should be considered the least susceptible to infection with this pathogen, while the cultivar Romulus as the most susceptible.

The presence of *F. equiseti* on oat panicles posed a risk of grain contamination with toxic metabolites of type A and B trichothecenes.

The differential response of the studied oat genotypes to *F. equiseti* infection confirmed that the selection of cultivars characterized by the lowest susceptibility to *Fusarium* spp. can be an effective method of reducing yield losses and grain contamination with mycotoxins.

**Supplementary Materials:** The following supporting information can be downloaded at: https://www.mdpi.com/article/10.3390/agronomy12020295/s1, Table S1: Effect of inoculation of oat panicles with *F. equiseti* No O-020 on the number of kernels per panicle, kernels yield per 10 panicles and 1000 kernels weight, average in 2015; Table S2: Effect of inoculation of oat panicles with *F. equiseti* No O-020 on the number of kernels per panicle, kernels yield per 10 panicles and 1000 kernels weight, average in 2016; Table S3: Effect of inoculation of oat panicles with *F. equiseti* No O-020 on the number of kernels per panicle, kernels yield per 10 panicles and 1000 kernels weight, average in 2017.

**Author Contributions:** Conceptualization, E.M., E.P., M.C., M.W. and W.W.; methodology, E.M., M.W., W.W., E.P., M.C.; software, E.M., M.W. and W.W.; validation, E.M., M.C, W.W. and M.W.; formal analysis and software, E.M., M.W.; investigation, E.M., M.C. and M.W.; resources, E.M.; data curation, E.M. and M.C.; writing—original draft preparation, E.M. and E.P.; writing—review and editing, E.M., E.P., M.C., W.W. and M.W.; visualization, E.M., E.P., W.W. and M.W.; supervision, E.M.; project administration, E.M.; funding acquisition, E.M. and M.C. All authors have read and agreed to the published version of the manuscript.

**Funding:** This research was financed by the Polish Ministry of Agriculture and Rural Development OKF/MR/24/2015–2017- HOR hn-801-PB-9/15; HOR hn-801-PB-7/16; HOR.hn.802.22.2017.

**Data Availability Statement:** The data presented in this scientific article are available from the corresponding author.

**Conflicts of Interest:** The authors declare no conflict of interest.

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
