# Peer review of "Reaction of Oat Genotypes to Fusarium equiseti (Corda) Sacc. Infection and Mycotoxin Concentrations in Grain"

_agronomy, doi:10.3390/agronomy12020295_

Round 1

Reviewer 1 Report

The article “Reaction of Oat Genotypes to Fusarium equiseti (Corda) Sacc. 2 Infection and Mycotoxins Concentrations in Grain” is of agronomic relevance. As general comment the work is well written and designed with relevant results. I have some minor points:

Introduction

In addition to the use of more resistant genotypes, what other measures are being used in the integrated control against Fusarium spp.? Given the importance of cultivation for human consumption, it would be worth extending this part a bit more.

Material and Methods

How did you get the isolate?

How was made the re-isolation of the fungus?

Results

Please a figure with healthy panicles and those showing disease symptoms should be provided.

Again, re-isolation of the fungus procedure and results should be better explained.

Table 1

Replace F.eq. by: “Inoculated” or “F.eq. inoculated”

Discussion

Line 252 F. equiseti in cursive

Author Response

Dear Reviewer,

thank you for your review.

I am sending an answer to your suggestions in the file

Reviewer 2 Report

In this manuscript, the authors attempted to describe the reaction of oat genotypes to Fusarium equiseti (Corda) Sacc. infection and mycotoxins concentrations in grain. The F. equiseti No. O-020 was used to invest its effect on 10 oat genotypes over three years (2015, 2016, and 2017) on the field experiments. However, the obtained data in this current manuscript were not robust and incomprehensive. Detailed information on the experimental design was lacking. The usage of English and written expression in this manuscript were unsatisfactory for publishable standards. This reviewer thus suggests having a substantial revision of the study before further consideration.

Here are major comments:

  1. All information related to collection, isolation, identification, and morphology of F. equiseti No. O-020 was lacking in the current manuscript, thus it is impossible to track if this strain is suitable to use as model research.
  2. More information related to FBH dedicatedly caused by F. equiseti should be added to specify the rationale of this study.
  3. Please add the description for detailed information related to not only the precisely conducted time points 2015, 2016, and 2017 but also experimental procedures. How many replicates of experiments have been done for each year and for a total of the 3-year period? How were plants cultivated in the 10 m2 each? How long did each experiment take from infection to harvest samples? How many samples were harvested?
  4. Please add a specific description for biochemical analyses, including what materials were used? How to prepare the materials? How many grams were used to analyze? Etc.
  5. Please add representative photos for the data demonstrating virulence, grain yield, and 1000 kernel weight.
  6. Except for table 1, where were the results for control experiments? In addition, how to obtain data in figures 1, 2, 3? What were the mean data in Figures 1, 2, 3?
  7. Line 118-124: where was the data for these statements?
  8. Line 168-169: where was the data for this statement? Also, which methods were used to isolate and identify the infected fungi?
  9. Please add a discussion of the relationship between weather conditions and FHB caused by F. equiseti No. O-020.
  10. The use of English in this manuscript is unsatisfying and it has so many typos and grammar errors throughout the manuscript. Therefore, it must be run by a native speaker in order to meet the professional standard for publishing as an original research article.

Author Response

(The authors gave the same response as above.)

Reviewer 3 Report

In this manuscript the authors describe experiments to evaluate 10 cultivars of oat for susceptibility to Fusarium equiseti and production of mycotoxins in the infested grain. Generally, the manuscript is clearly written and conclusions are based on the results obtained. The information will be useful for oat breeders, oat farmers, and food processors. My primary concern is the way in which the results are presented. I think the manuscript would be improved by having the authors consult a statistician. The first results that are presented are kernel number, grain yield and 1000 kernel weights averaged over the three years of the experiment. There was no statistical evidence presented that it was legitimate to combine results across years. In fact, in the Discussion the authors say there was high variability across years (line 271-274). I agree with the authors that average data is important so that the cultivar that performs well under different environmental conditions can be identified. However, readers need to understand that there was a significant environmental interaction. An ANOVA table showing means square results as the first data table would allow the authors to discuss the significant variables in the Results section where it belongs. ANOVA was used for analyzing all data. However, percent reduction data shown in Figures 1-3 are not normally distributed data and need to be analyzed with nonparametric methods. These data should be re-analyzed appropriately. The means presented in Figures 1-3 are a bit difficult to decipher since they are associated with the 2016 bar. Perhaps those data can be placed in a separate table or figure. The yield data in the paper are reported to two decimal places and the mycotoxins to 4 decimal places. Are these significant figures? Could each variable be measured that precisely or are the decimal places due to averaging? Please review these data for significant figures and edit as needed.

Additional specific concerns:

Line 75: How was strain No. O-020 identified? F. equiseti is generally considered part of a species complex. See this website for resources to identify Fusarium species http://isolate.fusariumdb.org/blast.php. More information on the source of this strain and its identification should be included. Why was this strain selected for use in the experiment? Is it deposited in a culture collection for others to use? Additionally, what was the source of the seed used? How could someone obtain these varieties? What is meant by a strict field experiment?

Line 86: Did all oat lines flower at the same time? What type of foil was used? Aluminum foil?

Line 93: More details are needed on the extraction method. Is there a reference for this? What was the sample size, number of replications? Was grain ground? What was the type of solid phase column?

Line 132: In the table legend please indicate the number of panicles used for measuring the number of kernels. Suggest …average from three study years.

Figures 1, 2,3. Need to add y-axis and x-axis labels.

Line 168: Delete carried out according to Koch’s postulates. Add a section in the Materials and Methods on how the fungus was re-isolated and identified.

Line 203: The first paragraph of this section on weather conditions should be moved to the Materials and Methods.

Line 231: Need to be consistent in how the F. equiseti strain is identified (No. 20 or No. O-020). How was it determined that there was a lesser effect on kernel weight than on number of kernels?

Minor editorial issues:

Line 16: remove to

Line 21,22: check spelling of deoxynivalenol and nivalenol.

Line 26, 42: inter alia is not in common usage in English. Suggest rewording to replace this term.

Line 92: Suggest …conducted to complete Koch’s postulates.

Line 144: Define TKW.

Line 123: Suggest …with a layer of mycelium.

Line 124: remove combination.

Line 151: Suggest Reduction of kernel number after panicle inoculation…

Line 159: Refer to just Figure 3.

Table 2, 4: Suggest: Average mycotoxin content over 3 years.

Table 3: Check column headings. What is toksyna?

Table 6: Suggest growth instead of vegetation. Rainfall not rainfalls.

Line 226: Need a period after [37].

Line 263: Change to mg/kg to match tables.

Author Response

(The authors gave the same response as above.)

Round 2

Reviewer 2 Report

Thank the authors for revising the manuscript to make it meet publishable quality.

Author Response

I would like to thank the Reviewer for their valuable comments which significantly contributed to the improvement of the quality of our manuscript.

The manuscript was re-examined and revised. The changes suggested by the Reviewers (Round 2) were introduced. A moderate English changes were also performed.

Reviewer 3 Report

In the revision the authors have addressed my concerns and improved the manuscript. I have only minor suggestions for further improvement. Please see the suggestions on the attached pdf file. 

line 135: The medium for preparing the inoculum is described as using 1 part SNA. Is this Selective Nutrient Agar? It is not clear if the medium was a liquid or solid medium.

line 249: Here, and other places in the manuscript, be careful about the use of "after". If you are describing the average of data from the three years, you should say "over 3 years of research" not after 3 years of research. 

This sentence in the legends of Figures 3, 4, and 5 is confusing.  "The average values marked with the same letter do not differ significantly at P ≤ 0.05." The average yellow bars? The averages in the far right columns labeled Average? Were comparisons made just within a cultivar or with all cultivars? Or were comparisons made within a year for all cultivars? Please clarify the legends.

line 355: The sentence is confusing: 

This fungus was less unfavorable to spring wheat ears than F. avenaceum, F. culmorum and F. graminearum under controlled conditions [13]. 

What fungus? Are you referring to F. equiseti? Instead of less unfavorable, can you say less aggressive or less pathogenic?

Author Response

I would like to thank the Reviewer for their valuable comments which significantly contributed to the improvement of the quality of our manuscript.

Respons to Reviewer

Comments

1: line 135: The medium for preparing the inoculum is described as using 1 part SNA. Is this Selective Nutrient Agar? It is not clear if the medium was a liquid or solid medium.

Response 1: SNA liquid selective medium, i.e. without the addition of agar, was used to prepare the infection suspension. This information was provided in the manuscript.

2: line 249: Here, and other places in the manuscript, be careful about the use of "after". If you are describing the average of data from the three years, you should say "over 3 years of research" not after 3 years of research.

Response 2: The phrase "after three years" was corrected in the manuscript.

  1. This sentence in the legends of Figures 3, 4, and 5 is confusing. "The average values marked with the same letter do not differ significantly at P ≤ 0.05." The average yellow bars? The averages in the far right columns labeled Average? Were comparisons made just within a cultivar or with all cultivars? Or were comparisons made within a year for all cultivars? Please clarify the legends.

Response 3: Legends for Figures 3-5 was verified

The values obtained within an individual year for all genotypes, marked with the same letter do not differ significantly at P ≤ 0.05.

The average values obtained for all genotypes over the three years of study (yellow columns), marked with the same letter do not differ significantly at P ≤ 0.05.

  1. line 355: The sentence is confusing:

This fungus was less unfavorable to spring wheat ears than F. avenaceum, F. culmorum and F. graminearum under controlled conditions [13].

What fungus? Are you referring to F. equiseti? Instead of less unfavorable, can you say less aggressive or less pathogenic?

Response 4: Was verified: F. equiseti was less pathogenic to spring wheat ears than
F. avenaceum, F. culmorum and F. graminearum under controlled conditions [13].

The manuscript also includes changes marked by the Reviewer in the PDF file.